# Association between Mood and Sensation Seeking Following rTMS

**DOI:** 10.3390/brainsci13091265

**Published:** 2023-08-30

**Authors:** Ulrike Kumpf, Aldo Soldini, Gerrit Burkhardt, Lucia Bulubas, Esther Dechantsreiter, Julia Eder, Frank Padberg, Ulrich Palm

**Affiliations:** Department of Psychiatry and Psychotherapy, Ludwig Maximilians University Munich, Nussbaumstr. 7, 80336 Munich, Germany; ulrike.kumpf@med.uni-muenchen.de (U.K.);

**Keywords:** non-invasive brain stimulation, transcranial magnetic stimulation, personality trait

## Abstract

Previous studies investigating mood changes in healthy subjects after prefrontal repetitive transcranial magnetic stimulation (rTMS) have shown largely inconsistent results. This may be due to methodological issues, considerable inter-individual variation in prefrontal connectivity or other factors, e.g., personality traits. This pilot study investigates whether mood changes after rTMS are affected by personality parameters. In a randomized cross-over design, 17 healthy volunteers received three sessions of 1 Hz rTMS to Fz, F3 and T3 (10/20 system). The T3 electrode site served as the control condition with the coil angled 45° to the scalp. Subjective mood was rated at baseline and after each condition. Personality traits were assessed using the NEO Five-Factor Inventory (NEO-FFI) and the Sensation Seeking Scale (SSS). For all conditions, a significant association between mood changes towards a deterioration in mood and SSS scores was observed. There were no differences between conditions and no correlations between mood changes and NEO-FFI. The data show that sensation-seeking personality has an impact on subjective mood changes following prefrontal rTMS in all conditions. Future studies investigating the effects of rTMS on emotional paradigms should include individual measures of sensation-seeking personality. The pre-selection of subjects according to personality criteria may reduce the variability in results.

## 1. Introduction

Non-invasive brain stimulation techniques like repetitive transcranial magnetic stimulation (rTMS) are used in clinical trials and in clinical practice for the treatment of mood disorders like major depressive disorder (MDD) [1,2,3]. To understand the mechanism of action of rTMS as a therapeutic intervention for mood disorders it is important to investigate the structure and neurobiological bases of affect. Study results in healthy subjects show a relationship between cortical excitability and mood [4] and further show that rTMS can modulate cortical excitability and thereby also influence network connectivity [5]. Early trials using high-frequency rTMS to investigate hemispheric language dominance have found an effect on mood in healthy subjects following rTMS over dominant frontal areas [6,7,8]. Subjects have experienced sadness and frustration and clinically observable mood change has been reported. Studies investigating mood changes after rTMS in healthy subjects have chosen the prefrontal cortex as the target area in the majority of reported data, as it is one of the cortical hub regions for mood generation and modulation. The results of former studies have demonstrated significant effects of prefrontal rTMS on mood in healthy subjects using a visual analogue scale (VAS) for mood rating. Left prefrontal rTMS led to a significant increase in the sadness ratings and a significant decrease in the happiness ratings compared to right prefrontal stimulation [6,9]. Another study also demonstrated a strong impact of rTMS on mood using the Beck Depression Inventory (BDI) in healthy subjects for mood rating [10]. Although it is highly questionable if the studies can be compared because of the use of different mood rating constructs and rTMS protocols, the finding that left prefrontal rTMS results in a significant reduction in the BDI indicates a contrary effect compared to the lateralized effect of the former studies. While 2 other studies found at least partial effects of prefrontal rTMS on mood in healthy subjects [11,12] and another study demonstrated that one intermittent Theta Burst Stimulation (iTBS) session can increase positive affective processing in healthy individuals [13], 11 other studies failed to demonstrate any mood effects in healthy subjects following prefrontal rTMS [14,15,16,17,18,19,20,21,22,23,24], iTBS or cTBS [25,26,27] even after 10 iTBS sessions [27]. In summary, the data on mood changes in healthy participants after rTMS is inconsistent for both older protocols (HF rTMS or LF rTMS) as well as for TBS (cTBS or iTBS), with the number of studies yielding negative results being predominant. This also applies to iTBS (one positive versus three negative studies).

A recent focus of rTMS research has been laid on the individual variability of responses across different system levels: motor cortex, prefrontal and other non-motor regions. For non-motor regions, the prefrontal cortex (PFC) is particularly relevant for therapeutic applications in affective disorders [28]. The prefrontal cortex (PFC) is a key region of the brain involved in mood regulation, emotional processing, executive functions, decision-making and personality expression. It is well-connected to other brain regions and forms complex networks that contribute to various aspects of human behaviour, including personality traits. Thus, PFC connectivity is related to individual factors, i.e., personality traits [29]. This association is shown for PFC connectivity and risk-taking behaviour [30] as well as sensation-seeking personality [31] and NEO Five-Factor Inventory (NEO-FFI) traits [32]. Furthermore, the connectivity of prefrontal areas and areas involved in the reward system is directly associated with mood and anxiety symptoms [33]. Embedded within these processes are also alterations in neurotransmitter concentrations and release (primarily dopamine) as additional links between personality factors, TMS effects and the condition of the prefrontal cortex [34,35]. Only a few studies involve TMS in addressing the association between prefrontal connectivity and personality traits. A TMS-EEG trial investigated the relationship between prefrontal interhemispheric connectivity and personality features as indexed by the NEO-FFI in healthy subjects. The results demonstrate that “agreeableness” as one of the measured personality traits correlates with prefrontal interhemispheric connectivity between the left and right dorsolateral prefrontal cortex (DLPFC) [36]. Another study found an association between higher scores in the personality trait “cooperativeness” and decreased cortisol output after active iTBS, but not after sham stimulation, when applied after a social stressor [26]. Taken together, the study results demonstrate that functional connectivity of prefrontal brain regions is associated with personality traits (especially sensation seeking) and risk-taking behaviour.

Despite these findings, studies investigating mood changes in healthy subjects after prefrontal repetitive transcranial stimulation (rTMS) have shown largely inconsistent results (see Appendix A). This may be due to methodological issues with rTMS, considerable inter-individual variation in prefrontal connectivity or other factors biasing this paradigm. The question of possible factors that influence the TMS effect with the perspective of a prediction of the response to TMS in the treatment of mood disorders points to personality factors with respect to the described relationships. This study addresses the issue of whether mood changes after 1 Hz rTMS are predominantly affected by personality parameters.

## 2. Materials and Methods

### 2.1. Subjects

Seventeen healthy right-handed volunteers (7 male) aged between 20 and 30 years (mean ± SD = 24.65 ± 3.74 years) were recruited for this pilot study by local advertisement. Announcements were posted in the Clinic for Psychiatry and Psychotherapy, as well as on the bulletin board of the Ludwig Maximilian University, Faculty of Medicine. These notices contained information about inclusion and exclusion criteria, the study procedure and compensation. Interested individuals could then contact the study team via email or phone. They gave their written informed consent after the procedure was fully explained. The exclusion criteria were current or past history of neurological or psychiatric disorder or neuropsychological performance below average. All subjects were naïve to TMS and received compensation (10 Euros per hour for the pre-assessment phase and 20 Euros per hour for their participation in the experiment). The experiment was conducted in accordance to the Declaration of Helsinki and local ethics board approval (Ludwig-Maximillian-University Munich, code number 229-98).

### 2.2. Trial Design

In a cross-over design, each subject received three sessions of rTMS on the three stimulation sites, medial PFC (mPFC), left DLPFC and auditory cortex (control condition). The wash-out period between the stimulations was one hour. Before undergoing the experimental procedure, resting motor threshold (rMT) was determined on a separate day. On the same day, participants performed the NEO Five Factors Inventory (NEO-FFI) personality test and the sensation seeking scale (SSS-V) for the assessment of individual personality traits. The stimulation procedure was executed within one day. After a baseline mood rating using fifteen items of the adjective word list globalform (EWL-G), three rTMS conditions were applied in a counterbalanced and randomized order. The participants relaxed between the stimulation conditions and after about 30 min after rTMS, the participants were asked to perform a mood rating again using the EWL-G. The experimental setup is shown in Figure 1.

### 2.3. Assessment of Personality Traits

Personality traits were assessed on a day prior to the experimental session. The subjects were asked to perform two self-rating personality tests, the NEO-Five Factor Inventory (NEO-FFI) by McCrae and Costa [37,38] and the Sensation Seeking Scale Form V (SSS-V) by Zuckerman. The NEO-FFI evaluates the five personality traits of neuroticism, extraversion, openness, agreeableness and conscientiousness. The test consists of 60 items and it needs about fifteen minutes to be completed. The test reliability depends on the measured personality factor between 0.71 and 0.85 [38]. The SSS-V evaluates the personality trait of sensation seeking in a four-dimensional model [39]. The sensation-seeking personality trait is described as the individual search for experiences and feelings that are novel, intense and complex. The four dimensions of sensation seeking are thrill- and adventure-seeking (TAS), experience-seeking (ES), disinhibition (DIS) and boredom susceptibility (BS). thrill- and adventure–seeking (TAS), experience-seeking (ES), disinhibition (DIS) and boredom susceptibility (BS). The validity of the four-dimensional model of sensation seeking is widely accepted [40]. The test consists of 40 items, each with two statements A and B, from which the test person can choose the most applicable one. The statements are assigned to one of the four dimensions of sensation seeking. A score for each of the four dimensions can be collected as well as the total value of sensation-seeking personality. The most recent version of the Sensation Seeking Scale (SSS-V) has demonstrated reasonable validity and test-retest reliability [41].

The reason for the selection of the two scales was to capture the personality traits that are associated with a change in mood in the sense of a specific response to a stimulus. It has been demonstrated that the factors “Extraversion” and “Neuroticism” are associated with positive and negative mood, respectively [42]. Depending on the individuals’ trait levels, they react differently to externally induced positive or negative moods [43]. However, a recent study failed to show the relationship between negative affect reactivity and the personality trait of neuroticism [44]. The personality trait “Sensation Seeking” was chosen because the experimental situation fulfils the conditions of providing a “sensation” for the participants. It is anticipated that the effect of this sensation is contingent with the individual’s level of sensation-seeking tendency [41]. Given the experimental setting, which presents an extraordinary situation for the participants, potentially acting as a stimulus for those with a stronger inclination towards sensation seeking, bodily sensations induced by TMS (e.g., during motor threshold determination) are also experienced. It is expected that the extent of the “Sensation Seeking” trait might influence the experience and, consequently, the effect of the stimulation in terms of mood alteration. It is anticipated that the degree of the sensation-seeking trait could potentially impact the encounter and, thereby, the impact of the stimulation on mood alteration. While the experimental context might evoke discomfort and unease in individuals with low sensation-seeking tendencies, those with a pronounced inclination for sensation seeking might perceive it as a stimulating and rewarding experience.

### 2.4. Mood Rating

Changes in affect were assessed by means of the multifactor instrument adjective checklist globalform (EWL-G, “Eigenschaftswörterliste”) by Janke and Debus [45]. The EWL-G can be used to capture relatively small, short-term changes in mood, which best meets the requirements in the experiments presented here. The subjects rated themselves before the stimulation (baseline T0) and about 30 min after each of the three 1 Hz-rTMS conditions (T1, T2, T3). On the 15 items of the EWL-G, each item with four adjectives is listed in a cloud expressing a specific mood state. On a Likert scale ranging from −10 to +10, the subjects indicated their agreement with or denial of the particular mood state. For the correlation analysis only, item 9 (good mood, happiness) and item 14 (unhappiness, depressed mood) of the EWL-G were evaluated. These two items reflect the mood states relevant to affective disorders, and comparability with other studies that also target mood change in these dimensions is ensured.

In most studies investigating mood changes in healthy participants, visual analogue scales (VAS) are employed for mood assessment. Questionnaires, such as the “Positive and Negative Affect Schedule” (PANAS), “Profile of Mood States” (POMS), or “Eigenschaftswörterliste” (EWL) used here offer better reliability than VAS [25]. Given its capacity to capture even minor, transient mood alterations, the EWL was chosen for the present study. The EWL is particularly suited for measuring effects resulting from key interventions such as environmental conditions (e.g., noise, temperature), therapeutic interventions (psychotherapy, pharmacotherapy), and interventions with motivational-emotional impacts within the realm of experimental psychology [45]. As such, the EWL serves as an exceptionally fitting instrument to capture mood changes following the rTMS interventions applied in this study.

### 2.5. Repetitive Transcranial Magnetic Stimulation Procedure

A Magstim rapid magnetic stimulator (Magstim Company Ltd., Whitland, UK) with a figure-8-shaped 70 mm coil was used for rTMS. The individual resting motor threshold was determined on a separate day prior to the experimental session. Motor evoked potentials (MEP) from the abductor pollicis brevis muscle (APB) were reported. The resting motor threshold (MT) was defined as the minimum stimulus intensity to evoke a MEP response of at least 50 µV from at least five out of ten consecutive trials. The stimulation sites were defined on the basis of the international 10/20 EEG system. The left DLPFC corresponded to the F3 and the mPFC to the Fz electrode site. The T3 electrode site targeting the auditory cortex served as the control condition, in lieu of a sham condition, with the coil angled 45° to the scalp. The centre of the coil was positioned over the cortical site (F3, Fz, T3 of the 10/20 system) in a frontal line with the handle pointing to the right hemisphere. The subjects received rTMS with an intensity of 120% of the individual MT and a frequency of 1 Hz for 10 min (600 pulses).

### 2.6. Statistics

The IBM SPSS statistics program (version 29) was used for statistical analysis. All datasets were tested for normal distribution and homogeneity of variances prior to analysis, and the appropriate test (parametric or non-parametric) was selected based on the results. The Kolmogorov–Smirnov test was conducted to assess normal distribution for each dataset. A non-significant result in the Kolmogorov–Smirnov test (*p* > 0.05) indicated normal distribution. Mauchly’s test of sphericity was performed to assess the homogeneity of variances. This test evaluates both the homogeneity of variances and the homogeneity of correlations over time. A non-significant test result (*p* > 0.05) indicated sufficient sphericity. In cases of non-homogeneity of variances (result in Mauchly’s test being significant with *p* ≤ 0.05), the Greenhouse–Geisser correction factor was applied to adjust the degrees of freedom. The correlation between subjective mood change after stimulation and the total value of the personality tests (SSS-V and NEO-FFI) was analysed using Pearson’s correlation coefficient. Pearson’s correlation was used due to its ability to quantify the strength and direction of linear relationships between continuous variables. This method provided valuable insights into the degree of association between mood changes and personality trait scores, allowing us to determine whether certain personality traits are more closely related to specific mood alterations after stimulation.

Mood change was defined as difference in the score of the EWL-G after stimulation (T1, T2, T3) compared to the baseline score (T0). Twelve tests for correlation between the parameters of personality and subjective mood change were executed. Additionally, the direct effect of rTMS on mood was analysed separately for the two items 9 and 14 of the EWL-G using one-way ANOVA including two (sham or active) and for another evaluation three (mPFC stimulation, left DLPFC stimulation, and sham stimulation) conditions.

The results of Pearson’s correlation analyses were interpreted as follows:


**Pearson’s r**

**Strength of Association/Correlation**
0None0 to ±0.25Negligible±0.25 to ±0.50Weak±0.50 to ±0.75Moderate±0.75 to ±1Strong±1Perfect

## 3. Results

All data were normally distributed according to the Kolmogorov–Smirnov test and met the criterion of homogeneity of variances according to Mauchly’s test. Repeated measures of ANOVA showed no significant difference in mood change in the tested groups (mPFC stimulation/left DLPFC stimulation/sham stimulation). ANOVA did not reject the null hypothesis of the two items of the EWL-G. Individual scores of EWL-9 and -14 for each subject are shown in Figure 2. Mean EWL scores for the 17 subjects after the different stimulation sessions and at baseline timepoint are shown in Figure 3.

Pearson’s correlation analyses revealed a moderate correlation between the absolute value of the SSS scores and mood change towards a deterioration in mood (a decrease in the EWL score in item 9—good mood, happiness) after each of the three conditions: mPFC (Fz): r = −0.683, *p* = 0.003, left dorsolateral PFC (F3): r = −0.580, *p* = 0.015, sham stimulation (T3): r = −0.523, *p* = 0.031 (Figure 4). The correlations between changes in mood in item 14 of the EWL (depressed mood) and SSS scores were consistently positive (indicating a deterioration in mood, as evidenced by increased agreement with this item). However, the *p*-values were consistently > 0.05 (Table 1). There were no correlations found for the NEO FFI factors.

## 4. Discussion

This study investigated the influence of individual personality traits on mood change after prefrontal rTMS in healthy volunteers in a sham controlled crossover design. Our results demonstrate a potential influencing variable to explain the inconsistent results in former studies addressing mood change in healthy subjects following prefrontal rTMS. The major finding of this study is that the parameter of individual sensation-seeking personality has an impact on mood change after 1 Hz rTMS in healthy subjects. Assuming that rTMS has an impact on mood per se, but that its direction and extent depends on the sensation-seeking personality of the subjects, the correlation found here could be explained as follows: The concept of the sensation-seeking personality factor is based on the theory that people are different in terms of their need for external stimuli [46]. This different level of need is determined by catecholamine metabolism in the brain. There is some evidence to underpin a relationship between sensation seeking, dopamine and rTMS.

Thus, the finding that mood change in healthy volunteers after prefrontal 1 Hz rTMS and sham rTMS was negatively correlated with sensation-seeking personality suggests that individuals with higher levels of sensation seeking may have a different response to rTMS compared to those with lower levels of sensation seeking. Specifically, individuals with higher sensation seeking scores may experience a greater decrease in mood following rTMS, regardless of whether they receive active or sham stimulation. This finding may be explained in part by the role of dopamine in the brain. Thus, sensation-seeking behaviour has been linked to differences in dopamine function, with higher levels of sensation seeking associated with increased dopamine release in response to rewarding stimuli [34,35,47]. RTMS has been shown to modulate dopamine release in various brain regions, including the prefrontal cortex, which has been implicated in regulating emotion and mood [48,49,50]. Therefore, it is possible that individuals with higher levels of sensation seeking have a more sensitive dopamine system, which may be more strongly affected by rTMS. Speer et al. found that 1 Hz rTMS inhibits metabolic processes in the brain [51] using (15) O water and positron emission tomography to measure changes in absolute regional cerebral blood flow. Furthermore, Shaul et al. were able to demonstrate that in cell cultures, rTMS with lower frequencies (3 Hz) caused a decrease in the release of norepinephrine and dopamine, while in contrast, higher-frequency rTMS (9 Hz) led to an increase in norepinephrine release [52]. Supporting this finding, it has also been shown repeatedly that high-frequency rTMS over PFC results in increased dopamine release [49]. If 1 Hz rTMS is able to decrease dopamine release in some brain regions, that could lead to a decline in mood ratings in individuals with higher levels of sensation seeking. Similarly, the placebo effect of sham rTMS may be more potent in individuals with higher sensation seeking, leading to a greater reduction in mood. However, this is just one possible explanation for the observed correlation between sensation seeking, mood change and rTMS, as the relationship between dopamine, sensation seeking and mood is complex and involves multiple brain regions and neurotransmitter systems.

Data show that sensation-seeking personality has a marked impact on subjective mood changes in volunteers following prefrontal rTMS. Interestingly, this was also the case for sham rTMS. That means for this study that subjective mood changes after rTMS resulted exclusively from the individual factor of sensation-seeking personality. The fact that no direct effect of active rTMS compared with sham rTMS on subjective mood change was identifiable and that mood change according to the sensation-seeking personality was observed also after the sham condition supports the conclusion that mood changes were not particularly caused by rTMS but instead the experimental procedure itself. Several studies showed an effect of rTMS on mood in healthy subjects [6,9,10]. Thus, the finding of this study demonstrating that the effect of rTMS on mood is exclusively caused by the factor of sensation-seeking personality should be interpreted carefully. In this study, sham rTMS had the same effect on mood as active rTMS according to the individual sensation-seeking personality.

The limitations of this study include the potential use of an active sham condition: Although sham stimulation was executed at a 45° angle over T3 (auditory cortex) to prevent prefrontal stimulation, it could be possible that a partially active placebo stimulation was performed, because the determining criteria for an ideal sham rTMS condition are not yet sound [53]. The potential use of an active placebo could be one reason why no significant difference in mood change was found after sham rTMS and active rTMS. Previous studies used a variety of different sham conditions. Barrett et al. [12] used a sham coil, while other investigators angled the coil for the sham condition but most of them angled it 90° to the scalp (e.g., [15,19]) whereas other studies chose a 45° angle for the sham condition as in this study. Some previous studies did not perform any sham condition [9,22]. Another limitation could be the choice of the 1 Hz protocol: Previous studies used different stimulation protocols varying in intensity, frequency, number of trains, sessions and stimuli, duration of the intertrain interval and number of pulses per site. Most notably, this study used 1 Hz low-frequency rTMS, whereas nearly all of the other studies addressing the same issue, especially the studies revealing significant effects of rTMS on mood in healthy subjects [6,9], used high-frequency rTMS (>1 Hz). Only three other studies used 1 Hz low-frequency rTMS [12,21,22]: Using 1 Hz rTMS, Jenkins et al. [22] and Grisaru et al. [21] failed to demonstrate any mood effect, while Barrett et al. [12] found an effect on mood indicated by the Positive and Negative Affect Schedule (PANAS) after 1 Hz rTMS. In the same study, Barrett et al. applied 10 Hz high-frequency rTMS over the left and right DLPFC in a control group and compared the two groups (1 Hz group *n* = 5 and 10 Hz group *n* = 5). While after 1 Hz, rTMS mood changes in the PANAS have been reported, there were no mood effects detectable with the PANAS after 10 Hz rTMS with an affect questionnaire. Different constructs to evaluate subjective mood changes could also cause the inconsistent results and, considering the comparability of the studies, the use of the EWL-G is a limitation. Most of the studies, i.e., those that were able to demonstrate a lateralized effect of prefrontal rTMS on mood, used a visual analogue scale (VAS) for mood rating. Others used the PANAS, a self-report questionnaire that assesses the presence and intensity of positive and negative emotions. PANAS is designed to measure emotional states and mood in individuals. George et al. [9] found a lateralized effect of rTMS on mood like the other two studies did, but VAS-assessed mood changes were not reported. Mood changes were found only with a modified version of the National Institute of Mental Health (NIMH) mood rating scale. Many other studies using the VAS for mood rating failed to demonstrate a mood effect of rTMS. Regarding the issue of the great number of studies with negative results using VAS, this study used the EWL-G by Janke and Debus to detect mood change, expecting that this self-rating construct is able to capture the small changes caused by rTMS more reliably than the VAS. Another limitation is the crossover design. In this study, three rTMS conditions (1 Hz rTMS over left DLPFC, mPFC and auditory cortex = sham condition) were executed consecutively in a single session during one day. Other studies with positive findings also stimulated more than one site per session. It is likely that carry-over effects could have influenced the results. In line with the question regarding potential carry-over effects, it is worth noting that the effects of excitatory or inhibitory protocols on the motor cortex do not persist beyond 60 min [54]. However, it is important to highlight that EEG-TMS trials demonstrated that the EEG-effects of TBS can endure for up to 90 min [55]. This variability in the persistence of effects reinforces the consideration of potential carryover effects and their impact on the experimental outcomes. In this study, a 60 min washout period between stimulations was employed in an effort to strike a compromise between avoiding carryover effects and ensuring the practical feasibility of conducting the study. The mood rating in this study was about 30 min after each of the three rTMS conditions. It is still unclear what is the ideal time interval between rTMS and mood rating. It is possible that a mood rating subsequent to the rTMS condition without any break leads to a detection of stronger direct effects of rTMS on mood, i.e., significant difference in mood effects between active and sham TMS, as aftereffects of a single and short rTMS stimulation may quickly vanish. Finally, it remains elusive if sensation-seeking behaviour has a sustained modulating effect on repeated sessions of rTMS during a treatment regimen for mood disorders.

The strengths of this study lie in the structured examination of potential TMS effects on mood in healthy participants, followed by the subsequent consideration of personality variables as explanations for the inconsistent effects. In this regard, the factor of sensation-seeking personality emerged as a significant influencing factor. This can be theoretically explained through relevant associations with dopamine release, expectation, and prefrontal activation, thus aligning consistently with current study results. Furthermore, the study’s findings contribute to the ongoing discussion on individual variability of responses to rTMS and offer a potential factor for predicting effects.

## 5. Conclusions

In conclusion, this study demonstrates a significant effect of sensation-seeking personality on mood change in healthy subjects following rTMS, whereas no immediate effect of rTMS on mood was detectable.

Future studies investigating the effects of rTMS on emotional measures and paradigms should further elucidate the finding that sensation-seeking personality has an influence on mood change after rTMS. The pre-selection of subjects according to personality criteria may reduce variation in responses and lead to more consistent findings.

## Figures and Tables

**Figure 1 brainsci-13-01265-f001:**
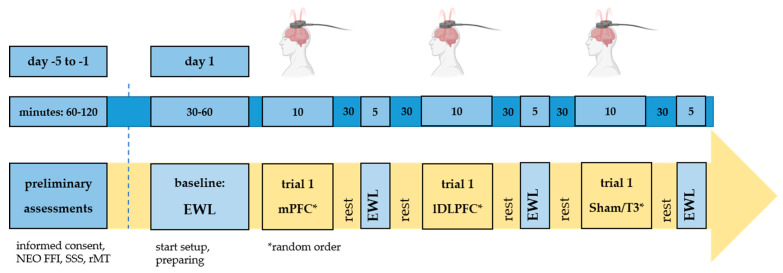
Design.

**Figure 2 brainsci-13-01265-f002:**
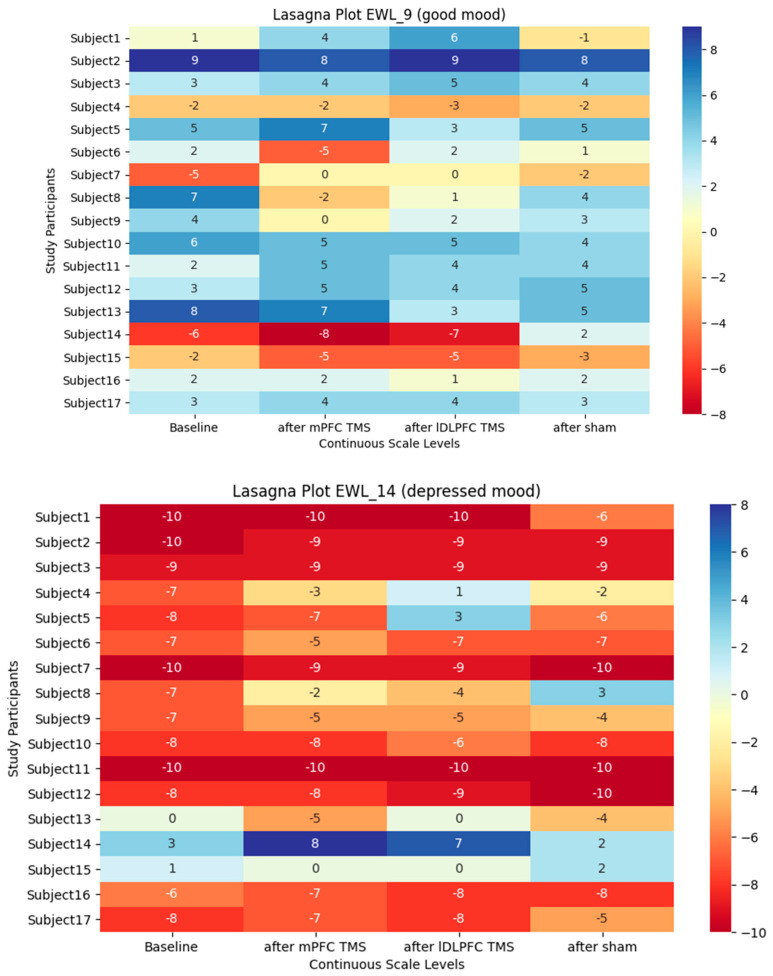
Lasagna plots show individual scores of EWL-9 and EWL-14 items at baseline and after the three TMS conditions.

**Figure 3 brainsci-13-01265-f003:**
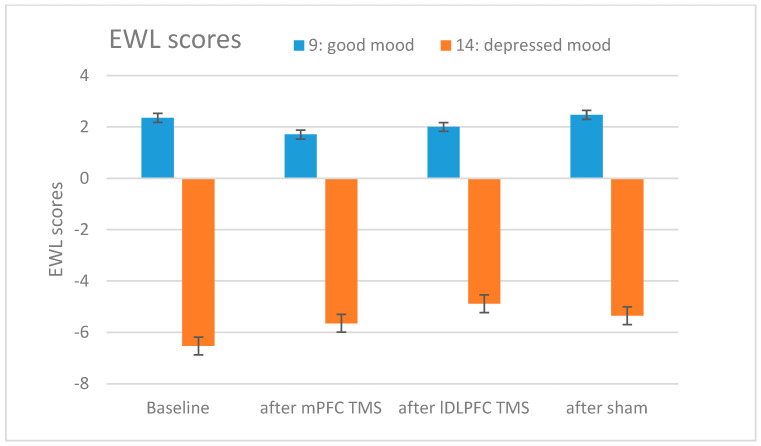
Mood rating scores in EWL-G, items 9 and 14 at baseline and mood measurement after the individual TMS. The graph shows mood rating scores as the mean of the 17 subjects with standard error.

**Figure 4 brainsci-13-01265-f004:**
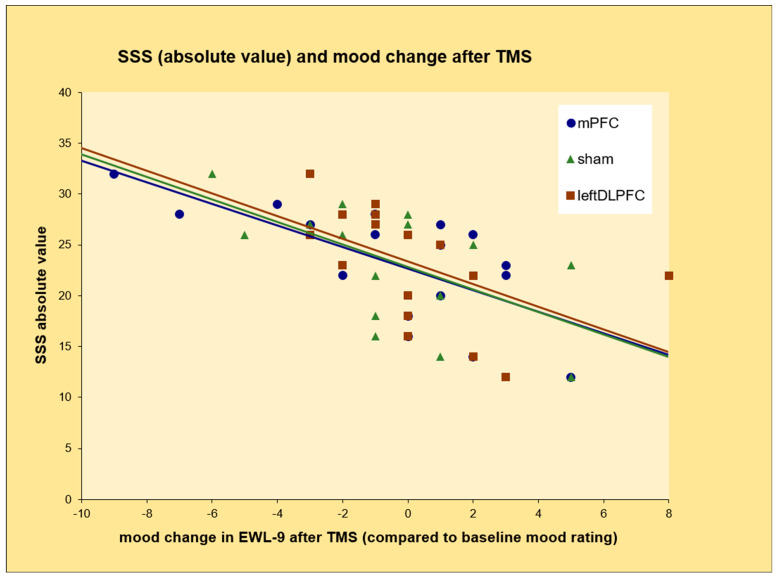
Correlation between total value of SSS-V and mood change after active and sham TMS. EWL-9: EWL-G, item 9.

**Table 1 brainsci-13-01265-t001:** Correlation grades between SSS Total Score and mood change in EWL-G, items 9 and 14, after TMS.

Pearson Correlation between SSS-V Total Score and Mood Change in EWL-9 (Good Mood) and EWL-14 (Depressed Mood) after TMS			
**Correlation Mood Change in EWL-9**	**Pearson’s r**	**Strength of Association/Correlation**	** *p* **
After lDLPFC TMS	−0.580	Moderate	0.015
After mPFC TMS	−0.683	Moderate	0.003
After sham TMS	−0.523	Moderate	0.031
**Correlation mood change in EWL-14**	**Pearson’s r**	**Strength of association/correlation**	** *p* **
After lDLPFC TMS	0.078	None	0.767
After mPFC TMS	0.044	Weak	0.867
After sham TMS	0.311	Weak	0.225

## Data Availability

The data presented in this study are available on request from the corresponding author. The data are not openly available due to privacy reasons.

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
