# Peer review of "Association between Mood and Sensation Seeking Following rTMS"

_brainsci, 2023, doi:10.3390/brainsci13091265_

Round 1

Reviewer 1 Report

The most interesting aspect of your paper was appendix A. The conclusions would be augmented be augmented by this paper’s results, showing that mood changes are not a TMS effect but a personality effect. I recommend you rewrite your title, simplify your introduction, and in the discussion generalize your findings beyond just healthy subject. 

Reviewer 2 Report

Thank you very much for the opportunity to review this interesting and well-written manuscript with clear aims. The topic is important and interesting, trying to find solutions for variable responses to neuromodulation by rTMS, and previous literature is non-consistent. The authors found that sensation seeking personality has an effect on the 1Hz effect. Another result was the placebo effect on T3 with tilted coil. The study has nice design, although I wonder if 17 subjects is enough for statistical significance. Could the authors provide power calculations?

I have some comments on the manuscript:

Introduction: Previous literature is quite nicely and largely cited, and the tables are detailed. However, the text could be more fluent, and state for example the iTBS clearly. Also I wish the study results pointing to an association between changes in prefrontal activation and personality traits would be briefly stated at the end of Introduction.

Methods: This does not need to be specified, but how much were the volunteers paid for participation? What was the delay (range) between rMT determination and the rTMS trials? Wash-out period and its effect on the results is discussed, but how was this chosen, based on what? The timeline in figure 1 could also clearly show the duration of rTMS and other details. How long did the whole measurement take? Do you think this would affect the results, for example subjects becoming bored, tired or frustrated? Which correlation method was used in statistical analysis?

Results: Make sure the results are in the same order throughout the manuscript (overall effect and SSS having effect on the magnitude). Which one is the primary one?

Discussion: This would benefit from one more round of language editing and reorganization (one topic in one paragraph). The transmitters are not mentioned in the Introduction, maybe they should? Could you explain PANAS briefly for non-experts in psychology? Carry-over effects are mentioned, but it was a bit diffuse. Maybe strength and limitations could be on separate paragraph(s)? Can you add reference for aftereffects and timing during what rTMS stimulation quickly vanishes? In the conclusions, direct effect of rTMS on mood was mentioned. Is direct an optimal word for this?

References are numerous. Can you add the method used in [47] in text as well?

Figures: It was clear that item 9 was good mood, happiness whereas 14 was happiness, depressed mood, but one of these would be clearer in the figures. In Figure 3 this was named EWL-9. Figure 2 would be better if individual values instead of group means were shown. This would also show if there were some outliers in the data. The figure for good mood shows different direction of effect for sham. Are there any reasons or speculations for this?

Tables: The tables could be divided into several ones (high-frequency, iTBS), so 1Hz studies were separately, since it is the most important issue regarding this study. Also, write this clearly in the Discussion (lines 300-306 on page 8). Can you state ‘none’ if sham was not used. Conclusion (p<0.05) was a bit odd. Where there two tables (another one with three columns).

Language is good, however, past tense is not congruently used throughout the text. Use of Capital letter (mainly in tables) should be checked.

Reviewer 3 Report

It's counterintuitive to expect that rTMS would significantly impact personality. Typically, factors causing substantial changes in personality involve events like severe trauma (such as traumatic brain injury) or prolonged exposure to stimuli affecting the brain. From my perspective, rTMS seems to influence mood, especially in individuals with high levels of sensation seeking. As a result, the current title appears misleading. Consider revising it to something like "Association between Mood and Sensation Seeking following rTMS."

Participant Recruitment:

Regarding the ethical implications of inviting individuals to undergo rTMS, could you provide more clarity on the recruitment process? It would be helpful to understand the specific methods used for advertisement or distribution of informational material, such as flyers, to potential participants.

Assessment of Personality Traits:

Could you elaborate on the necessity for employing two measurements to assess personality traits? It's essential to clarify the distinctions between the scores provided by both measurements and which one was utilized for the study.

Mood Rating Scale:

When referring to the mood rating, was a visual analog scale (VAS) employed? To enhance our understanding, please provide additional details regarding the scale's structure and administration.

Test Reliability (Line 132):

Could you clarify whether the reference to test reliability (line 132) pertains to Cronbach's alpha? An explanation of this term and its relevance in the study context would be appreciated.

Sensation Seeking Scale (SSS-V):

In relation to line 144, is it accurate to state that SSS-V is the most recent version, given that the reference dates back to 2007? Clarification regarding any updates or modifications since the referenced version would be beneficial.

Results and Mood Scores:

In the results section, it would be valuable to provide both the mood scores before and after the rTMS intervention (Mean (SD)), in addition to reporting the mood change scores.

Figure 3:

For a comprehensive understanding, could you please include the slope coefficients of each regression line depicted in Figure 3?

Correlation in Table 1:

Is the correlation reported in Table 1 adjusted for sensation seeking? If so, could you clarify which measurement of sensation seeking was used for this adjustment?

Limitations:

Finally, it's recommended to include a section discussing the limitations of the study. This will provide readers with a more balanced perspective on the findings and their implications.

Round 2

Reviewer 3 Report

Thanks for revising your manuscript. I think it is in good shape. I have no further questions. Congrats!

Author Response

Thank for your favourable feedback.